# Macromolecules with Different Charges, Lengths, and Coordination Groups for the Coprecipitation Synthesis of Magnetic Iron Oxide Nanoparticles as *T*_1_ MRI Contrast Agents

**DOI:** 10.3390/nano9050699

**Published:** 2019-05-05

**Authors:** Cheng Tao, Yanan Chen, Danli Wang, Yu Cai, Qiang Zheng, Lu An, Jiaomin Lin, Qiwei Tian, Shiping Yang

**Affiliations:** The Key Laboratory of Resource Chemistry of the Ministry of Education, The Shanghai Key Laboratory of Rare Earth Functional Materials, and The Shanghai Municipal Education Committee Key Laboratory of Molecular Imaging Probes and Sensors, Shanghai Normal University, Shanghai 200234, China; 1000441586@smail.shnu.edu.cn (C.T.); 1000441335@smail.shnu.edu.cn (Y.C.); wdl2011520@163.com (D.W.); 1000442867@smail.shnu.edu.cn (Y.C.); diwuxiaoo@163.com (Q.Z.); anlu1987@shnu.edu.cn (L.A.); qiweitian@shnu.edu.cn (Q.T.)

**Keywords:** *T*_1_-weight contrasts, iron oxide, surface charge, coprecipitation synthesis, MRI

## Abstract

Considerable efforts have been focused on the exploitation of macromolecule ligands for synthesis of magnetic Fe_3_O_4_ nanoparticles as *T*_1_ magnetic resonance imaging (MRI) contrast agents, but studies that concern macromolecule ligands with different charges and coordination groups are still limited. Herein, we used poly(acrylic acid) (PAA), poly(allylamine hydrochloride) (PAH), and polyvinyl alcohol (PVA), which possess negative, positive and neutral charges with carboxylic acid, amino and hydroxyl groups respectively, as templates and stabilizers to fabricate Fe_3_O_4_ nanoparticles through coprecipitation reaction. The obtained Fe_3_O_4_-PAA, Fe_3_O_4_-PAH, and Fe_3_O_4_-PVA nanoparticles showed *T*_1_ contrast performance with *r*_1_ relaxivities of 23.4, 60.3, and 30.6 mM s^−1^ at 0.5 T (25 °C), and a *r*_2_/*r*_1_ ratio of 2.62, 3.82, and 7.26, respectively. The cell viability assay revealed that Fe_3_O_4_-PAA and Fe_3_O_4_-PVA exhibited good biocompatibility, while Fe_3_O_4_-PAH displayed high cytotoxicity. In vivo *T*_1_-weighted (1 T) mice showed that both Fe_3_O_4_-PAA and Fe_3_O_4_-PVA were able to display remarkably brighten the contrast enhancement for the mice tumor and kidney sites, but Fe_3_O_4_-PAA had better contrast performance. This work highlights that the macromolecule ligands play an important role in the biocompatibility and *T*_1_ contrast performance of magnetic Fe_3_O_4_ nanoparticles.

## 1. Introduction

In the past decade, the utilization of magnetic iron oxide (Fe_3_O_4_) nanoparticles as a *T*_1_-weighted contrast for magnetic resonance imaging (MRI) has received tremendous attention [1,2,3,4,5,6]. On the one hand, Fe_3_O_4_ nanoparticles have better biocompatibility compared to currently widely-used clinical Gd-chelate *T*_1_ contrast agents, such as Gd-DTPA and Gd-DTOA [7,8,9,10,11,12]. The Gd-chelates have great potential accumulative toxicity (e.g., nephrogenic systemic fibrosis) caused by the leaching out of the Gd ions from the chelate ligand [13], while the Fe_3_O_4_ nanoparticles can be degraded in the body and the released iron ions can be subsequently incorporated into iron pools and metabolic processes [10]. On the other hand, Fe_3_O_4_ nanoparticles with a small size and suitable surface state are able to display a high longitudinal relaxation rate (*r*_1_) [14,15,16,17,18], which can significantly improve the spatial resolution of the *T*_1_-weighted image for some special sites such as blood vessels and vascular organs. Nevertheless, small Fe_3_O_4_ nanoparticles without appropriate ligands decorated on the surface tend to form aggregation and subsequently display *T*_2_ rather than *T*_1_ contrast enhancement [19,20].

To improve the *T*_1_ contrast performance and prevent the aggregation of nanoparticles in vivo, great efforts have recently been focused on the synthesis and surface modification of Fe_3_O_4_ nanoparticles [21,22,23,24,25]. The coprecipitation reaction of Fe^2+^ and Fe^3+^ ions under alkaline conditions is a traditional and widely-used method to fabricate Fe_3_O_4_ nanoparticles [26,27,28,29,30]. Compared with some other methods, such as thermal decomposition [31,32,33] and hydrothermal and solvothermal methods [34,35,36,37], coprecipitation is more convenient since it can be carried out in water phase without the further requirement of surfactant modification to improve the water-dispersibility of the obtained nanoparticles. Besides, similar to the approach of microwave-assisted synthesis [6,38], the coprecipitation method is a procedure that can be easily scaled up [39]. However, because the reaction is carried out in water phase and the reaction speed is very fast for coprecipitation, controlling the size and preventing the aggregation of the produced Fe_3_O_4_ nanoparticles are always important and challenging issues [40]. Using functional macromolecule ligands as templates and stabilizers has proved to be an effective approach to overcome these challenges [37,39,40,41,42]. The affinity coordination groups, such as the hydroxyl and carboxylic acid groups from the macromolecule ligands, can coordinate with the iron ions and thus control the growth of the nanoparticle seed and prevent the aggregation of the produced nanoparticles [43]. For example, Rui et al. used poly(acrylic acid) to synthesize Fe_3_O_4_ nanoparticles with a small size and high relaxivity for in vivo *T*_1_-weighted imaging [39]. Li et al. developed poly(acrylic acid)-poly(methacrylic acid) for the synthesis of small Fe_3_O_4_ nanoparticles with good water-dispersibility and remarkable *T*_1_ contrast performance [26].

To date, a great number of macromolecule ligands have been utilized as templates and stabilizers to fabricate hydrophilic small Fe_3_O_4_ nanoparticles as *T*_1_ contrast agents [26,27,44,45,46], but studies that concern the use of different charges and coordination groups of macromolecule ligands are still limited [47,48]. Indeed, the charges of macromolecule ligands are also an important parameter for controlling the size and preventing the aggregation of the Fe_3_O_4_ nanoparticles, since they usually correspond to the coordination affinity between the coordination groups and metal ions, and the electrostatic interaction between adjacent nanoparticles. In this work, we used three macromolecule ligands that possessed negative, positive and neutral charges with carboxylic acid, amino and hydroxyl groups, respectively, as templates and stabilizers for the coprecipitation synthesis of small magnetic Fe_3_O_4_ nanoparticles (Scheme 1), which showed differences in size, water-dispersibility, cytotoxicity and *T*_1_-weight contrast performance.

## 2. Materials and Methods

### 2.1. Materials

Fe(SO_4_)_2_·7H_2_O, FeCl_3_·6H_2_O and NH_3_ solution (25%) were obtained from Sigma Aldrich (Saint Louis, MO, USA). Poly(allylamine hydrochloride) (PAH, average Mw ~ 17,500), poly(acrylic acid) (PAA, average Mw ~ 2000) and polyvinyl alcohol (PVA, average Mw ~ 20,000–30,000) were purchased from Alfa (Heysham, UK). All chemicals were used without further purification.

### 2.2. Synthesis of Fe_3_O_4_-PAH, Fe_3_O_4_-PAA and Fe_3_O_4_-PVA Nanoparticles

The synthetic procedures for Fe_3_O_4_-PAH, Fe_3_O_4_-PAA and Fe_3_O_4_-PVA nanoparticles were quite similar, excepting the use of different polymer ligands as templates and stabilizers. Typically, the polymer ligand (140 mg) was added to a 250 mL three-necked flask with 50 mL of deionized water, and then stirred for 1 h under N_2_ atmosphere to remove the oxygen in the flask. Then 0.25 mmol of Fe(SO_4_)_2_·7H_2_O (70 mg) and 0.52 mmol of FeCl_3_·6H_2_O (140 mg) were dissolved in 2 mL of deionized water, and injected into the three-necked flask. The above mixture was slowly heated to 90 °C, and then 5 mL of concentrated ammonia solution was rapidly injected under vigorous stirring. The reaction was kept at 90 °C for a further 2 h and then cooled down to room temperature. The black suspension was ultrafiltration centrifugation (with 10-k ultra-filtration centrifuge tube) and was washed with deionized water 3–4 times.

### 2.3. Characterization

The structure of the obtained Fe_3_O_4_ nanoparticles was determined by Powder X-ray diffractometer (PXRD, Bruker, D8 ADVANCE, Cu K-α, Brucker, Karlsruhe, Germany). To verify the macromolecule ligand coating, Fourier-transform IR spectra (Nicolet Avatar 370 FT-IR, Thermo Electron Corporation, Madison, WI, USA) with potassium bromide as pressed pellets was carried out on a Nicolet Avatar 370 FT-IR spectrophotometer. The hydrodynamic size and zeta potential studies were carried out on a Malvern Zetasizer Nano ZS (Malvern, UK, scattering angle, 90°; temperature, 25 °C; refraction indexes of H_2_O and Fe_3_O_4_, 1.33 and 2.30, respectively). The morphology and size were evaluated using transmission electron microscopy (JEOL JEM-2010 microscopy, JEOL, Tokyo, Japan). The magnetic properties were measured using a superconducting quantum interference device (Lake Shore (Carson, CA, USA)). The concentration of nanoparticles was determined by dissolution in concentrated nitric acid (15 mol/L) and then ironion concentration was measured using high-dispersion inductively coupled plasma atomic emission spectroscopy (Teledyne Leeman Labs, Prodigy, Inc., Hudson, NH, USA). The longitudinal and transverse relaxation times (*T*_1_ and *T*_2_, respectively) for calculating the relaxation rate (*r*_1_ and *r*_2_, respectively) were measured on a magnetic resonance scanner (0.5 T, NMI20, Niumag, Shanghai, China) with parameters of DRG1, 3; TW, 8000 ms; RG, 20 db; SW, 100 kHz; SF, 18 MHz. *T*_1_-weighted images were obtained with the parameters of TR, 300 ms; TE, 0.04 ms; and *T*_2_-weighted images were acquired with parameters of TR, 4500 ms; TE, 100 ms.

### 2.4. In Vitro Cytotoxicity Assay

The cytotoxicity of Fe_3_O_4_-PAA, Fe_3_O_4_-PAH, and Fe_3_O_4_-PVA nanoparticles were evaluated using standard methyl thiazolyltetrazolium (MTT) assay with 4T1 cell lines (a mouse breast cancer cell line). The 4T1 cells were purchased from Shanghai Institutes for Biological Sciences. The MTT assay was carried out according to the following procedures. 4T1 cells (5 × 10^4^ cells/well) were first plated in a 96-well plate for 24 h and then treated with different concentrations (0, 5, 12.5, 25, 50, and 100 μg/mL) of Fe_3_O_4_-PAA, Fe_3_O_4_-PAH, or Fe_3_O_4_-PVA nanoparticles in dulbecco’s modified eagle medium (DMEM) for 12 or 24 h at 37 °C with 5% CO_2_. After that the medium was removed and the cells were washed with phosphate buffered saline (PBS), and then 20 μL (5 mg/mL) of thiazolyl blue tetrazolium bromide was added and the cells were incubated for a further 4 h. The medium was carefully removed and the remaining purple formazan crystals were lysed with 150 μL of dimethyl sulfoxide. The absorption of the formazan at 490 nm for calculating the cell viability was measured using a Multiskan MK3 microplate reader (Thermo Fisher Scientific, Waltham, MA, USA).

### 2.5. In Vivo Magnetic Resonance Imaging

To evaluate in vivo the MRI properties of Fe_3_O_4_-PAA and Fe_3_O_4_-PVA, *T*_1_-weighted images of mice bearing 4T1 tumor were carried out on a 1.0 T MRI scanner (NM42-040H-I; Niumag, Shanghai, China). The intravenous injecting dose of materials was 1.3 mg Fe/kg body weight. During imaging, the mice were anesthetized with 8% chloral hydrate. The *T*_1_-weighted imaging parameters were the following: field of view, 100 × 100 mm; slice thickness, 3 mm; echo time, 20 ms; repetition time (TR), 300 ms; matrix size, 256 × 192 mm. The operations of all animal methods were carried out strictly according to the requirements of the Animal Ethics Committee of the Shanghai Normal University (approval code No: 2018-0127).

## 3. Results and Discussion

### 3.1. Synthesis and Characterization

Giving that poly(acrylic acid) (PAA), poly(allylamine hydrochloride) (PAH), and polyvinyl alcohol (PVA) possess negative, positive and neutral charges with carboxylic acid, amino and hydroxyl coordination groups respectively, we used them as templates and stabilizers to fabricate Fe_3_O_4_ nanoparticles (denoted as Fe_3_O_4_-PAA, Fe_3_O_4_-PAH, and Fe_3_O_4_-PVA, respectively) through the coprecipitation reaction [49]. Typically, FeCl_3_, Fe(SO_4_)_2_ and PAA/PAH/PVA were mixed in deionized water and heated to 90 °C under a nitrogen atmosphere. Concentrated ammonia solution (25%) that served as the alkali was injected into the mixture to trigger the formation of Fe_3_O_4_ nanoparticles. In the beginning, the aqueous solution of iron ions and PAA, PAH or PVA were tawny and opaque. The pH of PAA, PAH and PVA in aqueous solution was 2.9, 5.8 and 3.8, respectively, and upon addition of ferric/ferrous salt, the corresponding pH changed to 1.5, 2.2 and 2.0, respectively. After the addition of concentrated ammonia solution, the reaction mixture became black immediately, suggested the formation of Fe_3_O_4_ nanoparticles.

Transmission electron microscopy (TEM) images showed that the average diameters of Fe_3_O_4_-PAA, Fe_3_O_4_-PAH, and Fe_3_O_4_-PVA nanoparticles were 4.5 ± 0.8 and 7.4 ± 1.3 and 2.8 ± 0.4 nm, respectively (Appendix A). The slightly different sizes of the nanoparticles indicated the different capabilities of PAA, PAH and PVA as templates and stabilizers in controlling the growth of the Fe_3_O_4_ seeds. The crystalline structures and phase composition of the as-synthesized nanoparticles were characterized via X-ray diffraction (XRD). As shown in Figure 1d, Fe_3_O_4_-PAA, Fe_3_O_4_-PAH, and Fe_3_O_4_-PVA have similar diffraction peaks, suggesting that they are all crystalline, which is important for magnetic nanoparticles. The 2-theta peaks at 30.4, 35.8, 43.5, 53.9, 57.5 and 63.2° were able to be indexed to {311}, {222}, {400}, {422}, {511} and {440} lattice planes of the cubic phase Fe_3_O_4_ (JCPDS No. 75-1449). No other peaks that belonged to this phase were observed for the three samples, confirming the successful fabrication of the pure phase Fe_3_O_4_. Fe_3_O_4_-PAA, Fe_3_O_4_-PAH, and Fe_3_O_4_-PVA exhibited good solubility in the aqueous solution (Figure 1e). The hydrodynamic size of Fe_3_O_4_-PAA, Fe_3_O_4_-PAH, and Fe_3_O_4_-PVA determined by dynamic light scattering (DLS) were around 56, 229 and 141 nm, respectively (Appendix A). Compared to the size observed from the TEM images, the larger hydrodynamic size of Fe_3_O_4_-PAA can be attributed to the surrounding macromolecule ligands and water molecules on the surface of the nanoparticles. Nevertheless, the hydrodynamic size of Fe_3_O_4_-PAH and Fe_3_O_4_-PVA were obviously larger than that observed in TEM images, suggesting the slight aggregation of Fe_3_O_4_-PAH and Fe_3_O_4_-PVA in the aqueous solution. The zeta potential of Fe_3_O_4_-PAA, Fe_3_O_4_-PAH, and Fe_3_O_4_-PVA was determined to be −35.8, 39.0, and −1.1 mV, respectively (Figure 2b). The negative, positive and nearly neutral zeta potential of Fe_3_O_4_-PAA, Fe_3_O_4_-PAH, and Fe_3_O_4_-PVA were consistent with that of the PAA, PAH and PVA (−19.2, 25, and −0.83 mV, respectively), indicating the existence of macromolecule ligands on the surface of the nanoparticles.

The surface functionalization of the Fe_3_O_4_ nanoparticles with PAA, PAH and PVA ligands was further confirmed by Fourier transform infrared (FT-IR) spectrum. As shown in Figure 2a, Fe_3_O_4_-PAA, Fe_3_O_4_-PAH, and Fe_3_O_4_-PVA exhibited a characteristic absorption peak of around 564–610 cm^−1^, which arose from the vibration band of Fe–O, suggesting the formation of Fe_3_O_4_. For PAA and Fe_3_O_4_-PAA, the absorption bands around 3200–3600 cm^−1^ can be attributed to the O–H stretching vibrations. The characteristic absorption peaks at 1712 and 1401 cm^−1^ were due to the C=O stretching vibrations and C–O stretching vibrations of carboxylic acid groups in the PAA, respectively [39]. The characteristic absorption bands for PAH and Fe_3_O_4_-PAH were observed at 3402, 3023 and 1604 cm^−1^, and can be attributed to the N–H stretching and bending vibrations, respectively. The absorption peak at 1113 cm^−1^ was due to the C–N stretching vibration [50]. For PVA and Fe_3_O_4_-PVA, characteristic absorption peaks were observed around 3200–3600 cm^−1^ (O–H stretching vibrations), 2940 cm^−1^ (–CH_2_– symmetric vibrations), 1094 and 1438 cm^−1^ (C–O stretching vibrations) [51]. The characteristic absorption of PAA, PAH and PVA can be observed in that of Fe_3_O_4_-PAA, Fe_3_O_4_-PAH, and Fe_3_O_4_-PVA, respectively, demonstrating the presence of macromolecules with the nanoparticles.

To evaluate whether the obtained Fe_3_O_4_ nanoparticles were superparamagnetic, which is crucial for magnetic Fe_3_O_4_ to be used as a *T*_1_-weighted MRI contrast agent, the magnetization curves for Fe_3_O_4_-PAA, Fe_3_O_4_-PAH, and Fe_3_O_4_-PVA with a magnetic field up to 1.5 T were measured at 298 K. The saturation magnetization for Fe_3_O_4_-PAA, Fe_3_O_4_-PAH, and Fe_3_O_4_-PVA were determined to be 32.1, 55.2 and 32.5 emu/g, respectively (Figure 2c). The inductively coupled plasma-atomic emission spectrometry (ICP-AES) results revealed that the ratios of Fe_3_O_4_ to the total weight were 70.1% for Fe_3_O_4_-PAA, 73.6% for Fe_3_O_4_-PAH, and 44.6% for Fe_3_O_4_-PVA. Compared with Fe_3_O_4_-PAH, the Fe_3_O_4_-PAA and Fe_3_O_4_-PVA with smaller diameters exhibited lower saturation magnetization, which can be ascribed to the higher specific surface associated with spin-canting [52]. According to the hysteresis loops, the coercivity and remanence for Fe_3_O_4_-PAA, Fe_3_O_4_-PAH, and Fe_3_O_4_-PVA were all negligible at room temperature, indicating their superparamagnetic behavior.

### 3.2. Magnetic Resonance Imaging

To assess the MRI properties of Fe_3_O_4_-PAA, Fe_3_O_4_-PAH, and Fe_3_O_4_-PVA, the *T*_1_-weighted image and longitudinal and transverse relaxation time (*T*_1_ and *T*_2_, respectively) with different concentrations of materials in aqueous solution at 25 °C were studied using a 0.5 T MRI scanner. As shown in Figure 3d, with the increasing concentration of materials, all the *T*_1_-weighted images of Fe_3_O_4_-PAA, Fe_3_O_4_-PAH, and Fe_3_O_4_-PVA gradually brightened, indicating that they were able to exhibit *T*_1_-weighted contrast [45]. According to the slope of the relaxometric curves (Figure 3a–c), the longitudinal molar relaxivities (*r*_1_) for Fe_3_O_4_-PAA, Fe_3_O_4_-PAH, and Fe_3_O_4_-PVA were calculated to be 23.6, 60.8, and 30.9 mM s^−1^, respectively, which are relatively high values as compared to some reported relaxivities for Fe_3_O_4_ nanoparticles using similar synthesis methods [53]. The high *r*_1_ relaxivity of these nanoparticles should be attributed to their small size and high surface decorated with hydrophilic macromolecules, which can facilitate the water exchange rate between the surrounding bulk water molecules, and which coordinated with the Fe ions. The transverse molar relaxivities (*r*_2_) for Fe_3_O_4_-PAA, Fe_3_O_4_-PAH, and Fe_3_O_4_-PVA were determined to be 62.3, 232.3 and 224.5 mM s^−1^, respectively. Generally, the *r*_2_/*r*_1_ ratio is a crucial parameter for assessing the *T*_1_ performance of contrast agents. The *r*_2_/*r*_1_ ratio for Fe_3_O_4_-PAA, Fe_3_O_4_-PAH, and Fe_3_O_4_-PVA was calculated to be 2.64, 3.82 and 7.27, respectively. For Fe_3_O_4_-PAA, the low *r*_2_ and *r*_2_/*r*_1_ ratio indicated that it is adequate as a *T*_1_-weighted contrast agent.

### 3.3. In Vitro Cytotoxicity

Encouraged by the good *T*_1_ contrast performance of Fe_3_O_4_-PAA, Fe_3_O_4_-PAH, and Fe_3_O_4_-PVA, their biocompatibility was assessed before in vivo tests. The cytotoxicity of Fe_3_O_4_-PAA, Fe_3_O_4_-PAH, and Fe_3_O_4_-PVA were investigated using 4T1 cell lines with standard methyl thiazolyltetrazolium (MTT) assay. For Fe_3_O_4_-PAA and Fe_3_O_4_-PVA, no significant cytotoxicity was observed after incubation with a different concentration of nanoparticles (0–100 μg mL^−1^ based on Fe) for 12 and 24 h. For Fe_3_O_4_-PAH, a significant decrease in the cell viability was observed even with very low concentrations of material. As shown in Figure 4a,c, the 4T1 cell that incubated with Fe_3_O_4_-PAA and Fe_3_O_4_-PVA remained viable above 94% and 84%, respectively, when the concentration of Fe ions was up to 100 μg mL^−1^ and the incubation time was prolonged to 24 h, indicating lower cytotoxicity for Fe_3_O_4_-PAA and Fe_3_O_4_-PVA within the investigated concentration. Nevertheless, the cells that were incubated with Fe_3_O_4_-PAH had a viability of only about 8%, although the concentration of Fe ions was only 25 μg mL^−1^ and the incubation time was only 12 h, suggesting a very large cytotoxicity of Fe_3_O_4_-PAH (Figure 4b). Generally, the cytotoxicity of a nanomaterial is associated with the surface properties of the nanoparticles. For Fe_3_O_4_-PAA, the low cytotoxicity of our results was in agreement with those reported in the literature, which showed that PAA-coated magnetic nanoparticles with highly negative zeta potential have good biocompatibility and are not internalized by biological cells [54]. For Fe_3_O_4_-PAH, the large cytotoxicity may be attributed to the abundant amino groups that have a high pKa value (around 9) of the PAH ligand on the surface of the Fe_3_O_4_ nanoparticles [48]. Due to the large cytotoxicity of Fe_3_O_4_-PAH, only Fe_3_O_4_-PAA and Fe_3_O_4_-PVA were used for further in vivo *T*_1_ MRI investigation.

### 3.4. In Vivo Magnetic Resonance Imaging

The *T*_1_-weight images with 4T1 tumor-bearing mice as the model were studied on a 1 T MRI scanner. Before intravenous injection of the materials, *T*_1_-weight images of the coronal planes as control groups were acquired on the MRI scanner, and then the mice were intravenously injected with Fe_3_O_4_-PAA/Fe_3_O_4_-PVA with a dose of 1.3 mg Fe/kg body weight, and acquired the *T*_1_-weight at different time points. As shown in Figure 5a, compared with the control group, the mice tumor and kidney sites brightened after 30 min, and gradually brightened with increasing time, suggesting Fe_3_O_4_-PAA displayed *T*_1_-weight contrast enhancement in these sites. To quantify the contrast, the signal-to-noise ratio was calculated through analyzing the target sites and the normal tissues of the *T*_1_-weight image. As shown in Figure 5b,c, after intravenous injection of Fe_3_O_4_-PAA, the relative signal-to-noise ratio of the tumor site reached the maximum (increased to about 65%) at 160 min, and that of the kidney site reached the maximum (increased to about 49%) at 60 min and remained almost unchanged for about 100 min. The increased *T*_1_ signals at the tumor and kidney sites can be attributed to the accumulation of Fe_3_O_4_-PAA nanoparticles that instinctively featured *T*_1_ contrast enhancement at these sites. After 180 min, the relative signal-to-noise ratio of both the tumor and kidney sites decreased slowly, indicating the metabolism of the nanoparticles. These results demonstrated that Fe_3_O_4_-PAA should be a good *T*_1_-weight contrast agent for in vivo MR imaging.

Similar to Fe_3_O_4_-PAA, the *T*_1_-weight images of 4T1 tumor-bearing mice slightly brightened the contrast enhancement at the tumor and kidney sites after intravenous injection of Fe_3_O_4_-PVA for 40 min (Figure 6a). The *T*_1_-weight signal reached the increased maximum of about 53% at 120 min for the tumor site (Figure 6b), and 17% at 120 min for the kidney site (Figure 6c), indicating that the Fe_3_O_4_-PVA were slowly accumulated and displayed *T*_1_ contrast enhancement in these sites. After 180 min, the relative signal-to-noise ratio of both tumor and kidney sites decreased slowly, suggesting the slow metabolism of the Fe_3_O_4_-PVA nanoparticles. The brightened contrast of the mice tumor and kidney sites demonstrated that Fe_3_O_4_-PVA can also be used as *T*_1_-weight MRI contrast agent. Nevertheless, the increased *T*_1_-weight signals at the tumor and kidney sites after intravenous injection of Fe_3_O_4_-PVA were slightly weaker than that for Fe_3_O_4_-PAA, indicating that Fe_3_O_4_-PAA would be a better *T*_1_-weight contrast agent.

## 4. Conclusions

In summary, Fe_3_O_4_ nanoparticles with the surface modified by negative, positive and neutral macromolecule ligands of PAA, PAH and PVA, respectively, were synthesized using the coprecipitation reaction. The obtained Fe_3_O_4_-PAA, Fe_3_O_4_-PAH, and Fe_3_O_4_-PVA nanoparticles showed slight differences in size and water-dispersibility. Besides, Fe_3_O_4_ nanoparticles modified with PAA and PVA showed good biocompatibility, while those modified using PAH displayed high cytotoxicity during cell viability assay. In vitro and in vivo experiments demonstrated that both Fe_3_O_4_-PAA and Fe_3_O_4_-PVA are adequate as *T*_1_-weighted contrast agents, but Fe_3_O_4_-PAA exhibited a better *T*_1_ contrast performance. This work highlights that the macromolecule ligands for modifying the Fe_3_O_4_ nanoparticles greatly affect their biocompatibility and *T*_1_ contrast performance, which should be helpful for the design of functional ligands for developing Fe_3_O_4_ based *T*_1_ contrast agents.

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
