# Peer review of "Macromolecules with Different Charges, Lengths, and Coordination Groups for the Coprecipitation Synthesis of Magnetic Iron Oxide Nanoparticles as T1 MRI Contrast Agents"

_nanomaterials, 2019, doi:10.3390/nano9050699_

Reviewer 1 Report

The manuscript by Tao et al presents the synthesis of magnetite NPs by the coprecipitation route (in a hot-injection configuration, at 90°C) combined by their coating with three different hydrophilic polymers and their subsequent study as MRI contrast agents (CAs). The three polymers used (PAA, PAH and PVA) induce different Fe3O4 core and hydrodynamic sizes depending on their chelating properties and their net charges (respectively negative, positive and neutral). A cell cytotoxicity study enables eliminating the PAH-coated Fe3O4 NPs in subsequent pre-clinical study on mice. Biodistribution study on a low magnetic field MRI scanner concludes that the PAA-coated NPs have superior properties than PVA-coated ones as they lead to significant signal enhancement at the tumor site. Globally this study is interesting and well made. However the Reviewer has some concerns before the manuscript can be accepted for publication in Nanomaterials, as detailed in the following points.

1) Authors provide the r1 and r2/r1 relaxivity values in the abstract: they should also specific the B0=0.5 T used there, as comparison of relaxivities between various publications can be made only at a given magnetic field (and also temperature, that has to be specified if it is not the 37°C physiological value).

2) Besides references [11-15], other references that could be mentioned are those that showed that to prepare Fe3O4 NPs with low r2/r1 value that can be used as efficient T1 CAs, not only a small core diameter should be designed, but also an efficient repulsive and hydrophilic coating to maintain the magnetic NPs as individually dispersed NP suspension, even under a high magnetic field: Borase et al, Angew. Chem. Int. Ed. 2013, 52, 3164-3167 (with polysaccharide coating) and Hannecart et al, Nanoscale, 2015, 7, 3754-3767 (with thermosensitive polymer coating, to detect temperature variation from the T1 signal changes).

3) A relevant review paper on the templated synthesis of magnetic NPs that could be cited is Nguyen el al, Nanomaterials 2014. 4: 628-685, where it is mentioned in particular that polymers with multivalent complexation ability introduced during synthesis have a direct influence on core sizes (as it is found here also). 

4) Line 80: ‘carboxyl’ to be changed by ‘carboxylic acid’

5) Authors indicate the ‘tawny’ aspect of ferric/ferrous salt mixtures with polymers and attributed it to coordination bonds between cations and polymers. This can also be a signed of a beginning of precipitation of hydroxides or oxo-hydroxides (see  Ahn et al J. Phys. Chem. C 2012, 116, 6069-6076, a reference describing the steps of co-precipitation route). To be clearer to the readers, it is important that authors specify the pH values of the initial solutions (the three polymers alone, and their mixtures with iron salts).

6) TEM images show a clear variation of core diameters between the 3 polymers (PVA<PAA<PAH), the PAH-coated ones having the classical (polydisperse) diameter distribution centered around 6-7 nm. Bothe images and histograms are convincing, but a lower number of digits to the mean and standard deviations values should be provided (two digits are not significant).

7) Regarding the hydrodynamic sizes as measured by dynamic light scattering (DLS), the order is PVA<PAA<<PAH), showing partial aggregation in the last case. First of all, more details should be given about the DLS measurements: the scattering angle (90° and 173° are standard values for Malvern devices), and, more importantly, the refraction indexes used not only of the solvent, but also of the materials. If authors want to derive number-average values as plotted on Fig 1.e, it is mandatory to provide the real and imaginary part of the materials refraction index as it must be used to deduce the N-average distribution from the z-average (intensity) distribution, using the Mie theory of light scattering. Standard values used for iron oxide NPs are RI=2.3 and k=0.1 at 633 nm wavelength.

8) Moreover, to correctly interpret the influence of chemical nature of the polymers on the aggregation, the molar mass (Mn) and mass dispersity (Mw/Mn) of the chains should be provided (as longer chains can produce NP bridging, whereas shorter chains can favor stabilization as individually dispersed NPs). 

9) Line 99: ‘belonged’ to be changed by ‘belonging’

10) Authors provide convincing magnetization data, with a saturation magnetization that gets much lower for the lower diameters (3-4 nm) compared to larger ones (6-7), which can be ascribed to the higher specific surface associated to spin-canting, as shown in Demortière et al, Nanoscale, 2011, 3, 225. In addition, can the authors deduced the ratio of iron oxide to total weight from their measurements (iron being titrated independtly by ICP-AES)? How would it compare to thermogravimetric results (if the authors have access to a TGA instrument)?

11) Line 149: as said already in 1), relaxivities can only be compared at given field and temperature. Here the r2/r1=7.3 for Fe3O4-PVA sample would it be even higher at a field of 1.5 T or even 3T, as used by clinical MRI scanners of most hospitals in the world.

12) Figure 3: it seems that the curve of 1/T1 or 1/T2 vs [Fe] go through the origin. Actually they should rather go through the 1/T1 or 1/T2 values of pure water. Can the authors provide those values?

13) Line 163: ‘well’ to be replaced by ‘good’

14) Regarding cytotoxic values, authors could mention the work by Safi et al in Nanotechnology 2010, 21, 145103 that shows that PAA-coated magnetic NPs with highly negative zeta potential are not internalized by biological cells (because of their negative membrane surface charge). Regarding PAA-coated samples, authors ascribe their high cytotoxicity to their positively charge: Actually this is only valid for –NH2 group that have high pKa value (around 9), because other week polycations like chitosan chains whose pKa is much lower (around 5-6) are non-toxic to cells.

15) Line 187 and Line 271: ‘Kg” to be replaced by ‘kg’

16) Line 243: ‘Hydration’ to be replaced by ‘hydrodynamic’

17) Line 247: ‘nitration’ to be replaced by ‘dissolution in concentrated nitric acid’ (specify concentration)

Author Response

Response to Reviewers

Reviewer 1:

Comment: The manuscript by Tao et al presents the synthesis of magnetite NPs by the coprecipitation route (in a hot-injection configuration, at 90°C) combined by their coating with three different hydrophilic polymers and their subsequent study as MRI contrast agents (CAs). The three polymers used (PAA, PAH and PVA) induce different Fe3O4 core and hydrodynamic sizes depending on their chelating properties and their net charges (respectively negative, positive and neutral). A cell cytotoxicity study enables eliminating the PAH-coated Fe3O4 NPs in subsequent pre-clinical study on mice. Biodistribution study on a low magnetic field MRI scanner concludes that the PAA-coated NPs have superior properties than PVA-coated ones as they lead to significant signal enhancement at the tumor site. Globally this study is interesting and well made. However the Reviewer has some concerns before the manuscript can be accepted for publication in Nanomaterials, as detailed in the following points.

Response: Thanks very much for the reviewer’s positive comments and constructive suggestions.

Comment: 1) Authors provide the r1 and r2/r1 relaxivity values in the abstract: they should also specific the B0=0.5 T used there, as comparison of relaxivities between various publications can be made only at a given magnetic field (and also temperature, that has to be specified if it is not the 37°C physiological value).

Response: Thanks very much for the reviewer’s remind. The relaxivities were measured on a 0.5 T MRI scanner at 25 °C, which have been given in the abstract in this revised manuscript.

The obtained Fe3O4-PAA, Fe3O4-PAH, and Fe3O4-PVA nanoparticles showed well T1 contrast performances with r1 relaxivities of 23.4, 60.3, and 30.6 mM s−1 at 0.5 T (25 °C), and r2/r1 ratio of 2.62, 3.82, and 7.26, respectively.

Comment: 2) Besides references [11-15], other references that could be mentioned are those that showed that to prepare Fe3O4 NPs with low r2/r1 value that can be used as efficient T1 CAs, not only a small core diameter should be designed, but also an efficient repulsive and hydrophilic coating to maintain the magnetic NPs as individually dispersed NP suspension, even under a high magnetic field: Borase et al, Angew. Chem. Int. Ed. 2013, 52, 3164-3167 (with polysaccharide coating) and Hannecart et al, Nanoscale, 2015, 7, 3754-3767 (with thermosensitive polymer coating, to detect temperature variation from the T1 signal changes).

Response: Thanks for the reviewer’s suggestion. These two references that prepared Fe3O4 NPs with low r2/r1 value and can be used as efficient T1 CAs have been cited in this revised manuscript as Ref. 17-18.

17.  Borase, T.; Ninjbadgar, T.; Kapetanakis, A.; Roche, S.; O'Connor, R.; Kerskens, C.; Heise, A.; Brougham, D. F. Stable aqueous dispersions of glycopeptide-grafted selectably functionalized magnetic nanoparticles. Angew. Chem. Int. Ed. 2013, 52, 3164-3167.

18.  Hannecart, A.; Stanicki, D.; Vander Elst, L.; Muller, R. N.; Lecommandoux, S.; Thévenot, J.; Bonduelle, C.; Trotier, A.; Massot, P.; Miraux, S.; Sandre, O.; Laurent, S. Nano-thermometers with thermo-sensitive polymer grafted USPIOs behaving as positive contrast agents in low-field MRI. Nanoscale 2015, 7, 3754-3767.

Comment: 3) A relevant review paper on the templated synthesis of magnetic NPs that could be cited is Nguyen el al, Nanomaterials 2014. 4: 628-685, where it is mentioned in particular that polymers with multivalent complexation ability introduced during synthesis have a direct influence on core sizes (as it is found here also). 

Response: Thanks for the reviewer’s kind suggestion. The reference reported by Nguyen el al, on Nanomaterials 2014, 4, 628-685 has been cited in this revised manuscript as Ref. 25.

25.  Nguyen, V. T. A.; Gauthier, M.; Sandre, O. Templated synthesis of magnetic nanoparticles through the self-assembly of polymers and surfactants. Nanomaterials 2014, 4, 628-685.

Comment: 4) Line 80: ‘carboxyl’ to be changed by ‘carboxylic acid’

Response: Thanks very much for the reviewer’s remind. The ‘carboxyl’ in line 80 has been changed by ‘carboxylic acid’ in this revised manuscript.

Comment: 5) Authors indicate the ‘tawny’ aspect of ferric/ferrous salt mixtures with polymers and attributed it to coordination bonds between cations and polymers. This can also be a signed of a beginning of precipitation of hydroxides or oxo-hydroxides (see Ahn et al J. Phys. Chem. C 2012, 116, 6069-6076, a reference describing the steps of co-precipitation route). To be clearer to the readers, it is important that authors specify the pH values of the initial solutions (the three polymers alone, and their mixtures with iron salts).

Response: Thanks for the reviewer. The pH for PAA, PAH and PVA in aqueous solution was determined to be 2.9, 5.8 and 3.8, respectively. Upon addition of ferric/ferrous salt, the corresponding pH was changed to 1.5, 2.2 and 2.0, respectively. In this revised manuscript, to avoid confuse for the readers, we changed the sentence of “In the beginning, the aqueous solution of iron ions and PAA, PAH or PVA were tawny and opaque, indicated that the iron ions interacted with the coordination groups from the macromolecule ligand.” as “In the beginning, the aqueous solution of iron ions and PAA, PAH or PVA were tawny and opaque. The pH of PAA, PAH and PVA in aqueous solution was 2.9, 5.8 and 3.8, respectively, and upon addition of ferric/ferrous salt, the corresponding pH was changed to 1.5, 2.2 and 2.0, respectively.”

In the beginning, the aqueous solution of iron ions and PAA, PAH or PVA were tawny and opaque. The pH of PAA, PAH and PVA in aqueous solution was 2.9, 5.8 and 3.8, respectively, and upon addition of ferric/ferrous salt, the corresponding pH was changed to 1.5, 2.2 and 2.0, respectively.

Comment: 6) TEM images show a clear variation of core diameters between the 3 polymers (PVA<PAA<PAH), the PAH-coated ones having the classical (polydisperse) diameter distribution centered around 6-7 nm. Bothe images and histograms are convincing, but a lower number of digits to the mean and standard deviations values should be provided (two digits are not significant).

Response: Thanks for the reviewer’s suggestion. A lower number of digits to the mean and standard deviations values have been provided in Figure S1 in this revised manuscript. The details are following.

Figure S1 Particle size distribution of (a) Fe3O4-PAA, (b) Fe3O4-PAH and (c) Fe3O4-PVA acquired from the TEM images.

Comment: 7) Regarding the hydrodynamic sizes as measured by dynamic light scattering (DLS), the order is PVA<PAA<<PAH), showing partial aggregation in the last case. First of all, more details should be given about the DLS measurements: the scattering angle (90° and 173° are standard values for Malvern devices), and, more importantly, the refraction indexes used not only of the solvent, but also of the materials. If authors want to derive number-average values as plotted on Fig 1.e, it is mandatory to provide the real and imaginary part of the materials refraction index as it must be used to deduce the N-average distribution from the z-average (intensity) distribution, using the Mie theory of light scattering. Standard values used for iron oxide NPs are RI=2.3 and k=0.1 at 633 nm wavelength.

Response: Thanks for the reviewer’s suggestion. More details parameters about the DLS measurements, including scattering angle, temperature, and refraction indexes of H2O and Fe3O4, have been provided in this revised manuscript. To avoid confuse to the readers, the hydrodynamic size profiles of Fe3O4-PAA, Fe3O4-PAH and Fe3O4-PVA nanoparticles using the form of the N-average distribution in Figure 1e have been removed, and that using the form of the z-average (intensity) distribution have been provided in Figure S2 in this revised manuscript. The details are following.

The hydrodynamic size and zeta potential studies were carried out on a Malvern Zetasizer Nano ZS (scattering angle, 90°; temperature, 25 oC; refraction indexes of H2O and Fe3O4, 1.33 and 2.30, respectively)

Figure S2. The hydrodynamic size profile of (a) Fe3O4-PAA, (b) Fe3O4-PAH and (c) Fe3O4-PVA nanoparticles in an aqueous solution.

Comment: 8) Moreover, to correctly interpret the influence of chemical nature of the polymers on the aggregation, the molar mass (Mn) and mass dispersity (Mw/Mn) of the chains should be provided (as longer chains can produce NP bridging, whereas shorter chains can favor stabilization as individually dispersed NPs).

Response: Thanks for the reviewer’s kind suggestion. We agree with the reviewer that the molar mass (Mn) and mass dispersity (Mw/Mn) of the chains are important parameters for correctly interpreting the influence of chemical nature of the polymers on the aggregation. The average Mw for PVA, PAA and PAH have been provided in the Material section this revised manuscript.

Poly(allylamine hydrochloride) (PAH, average M.w. ~17500), poly(acrylic acid) (PAA, average M.w. ~2000) and polyvinyl alcohol (PVA, average M.w. ~20000-30000) were purchased from Alfa. All chemicals were used without further purification.

Comment: 9) Line 99: ‘belonged’ to be changed by ‘belonging’

Response: Thanks very much for the reviewer’s kind remind. The ‘belonged’ in line 99 has been changed by ‘belonging’ in this revised manuscript.

Comment: 10) Authors provide convincing magnetization data, with a saturation magnetization that gets much lower for the lower diameters (3-4 nm) compared to larger ones (6-7), which can be ascribed to the higher specific surface associated to spin-canting, as shown in Demortière et al, Nanoscale, 2011, 3, 225. In addition, can the authors deduced the ratio of iron oxide to total weight from their measurements (iron being titrated independtly by ICP-AES)? How would it compare to thermogravimetric results (if the authors have access to a TGA instrument)?

Response: Thanks for the reviewer. We agree with the reviewer that the ratio of iron oxide to total weight is an important parameter for comparing the saturation magnetization of the Fe3O4-PAA, Fe3O4-PAH, and Fe3O4-PVA. Due to un-accessible TGA instrument in our laboratory, we only used ICP-AES to measure the ratio of iron oxide to total weight. For Fe3O4-PAA, Fe3O4-PAH, and Fe3O4-PVA, the ratio of Fe3O4 to total weight were determined to be 70.1%, 73.6%, and 44.6%, respectively.

ICP-AES results revealed the ratios of Fe3O4 to the total weight were 70.1% for Fe3O4-PAA, 73.6% for Fe3O4-PAH, and 44.6% for Fe3O4-PVA, respectively. Compared with Fe3O4-PAH, the Fe3O4-PAA and Fe3O4-PVA with smaller diameter exhibited lower saturation magnetization, which can be ascribed to the higher specific surface associated to spin-canting[52].

Comment: 11) Line 149: as said already in 1), relaxivities can only be compared at given field and temperature. Here the r2/r1=7.3 for Fe3O4-PVA sample would it be even higher at a field of 1.5 T or even 3T, as used by clinical MRI scanners of most hospitals in the world.

Response: Thanks for the reviewer’s kind remind. The magnetic field and temperature for the measurement of the relaxivities have been provided in this revised manuscript, and the relaxivityes were compared with that using the same measured magnetic field and temperature. According to the reference (Chem. Mater. 2018, 30, 7950−7961), the r2/r1 ratio would be higher at higher magnetic field. Due to the limited available devices and resources of our laboratory, it is challenge for us to measure the relaxivities at a higher field of 1.5 T or 3T. Nevertheless, according to the reviewer’s suggestion, we re-measured the in vivo MR imaging on a 1T MRI scanner. The details are following.

Figure 5. (a) T1-weighted MR images of mice collected before (control group) and after intravenous injection of Fe3O4-PAA at time points of 30, 60, 90, 120, 160 and 180 min. The corresponding relative T1-weighted signals extracted from (b) tumor (orange circle) and (c) kidney (dark yellow circle) sites.

Figure 6. (a) T1-weighted MR images of mice collected before (control group) and after intravenous injection of Fe3O4-PVA at time points of 40, 90, 120 and 180 min. The corresponding relative T1-weighted signals extracted from (b) tumor (orange circle) and (c) kidney (dark yellow circle) sites.

Comment: 12) Figure 3: it seems that the curve of 1/T1 or 1/T2 vs [Fe] go through the origin. Actually they should rather go through the 1/T1 or 1/T2 values of pure water. Can the authors provide those values?

Response: Thanks for the reviewer. We are very sorry for our mistakes. The curves of 1/T1 or 1/T2 vs [Fe] that go through the 1/T1 or 1/T2 values of pure water have been provide in Figure 3 in this revised manuscript. The details are following.

Figure 3. Plot of 1/T against Fe concentration for (a) Fe3O4-PAA, (b) Fe3O4-PAH, and (c) Fe3O4-PVA nanoparticles. The values of r1and r2 were calculated based on the slopes of the corresponding fitting lines. (d) T1-weight MR images of the aqueous dispersion of Fe3O4-PAA, Fe3O4-PAH, and Fe3O4-PVA nanoparticles with different Fe concentrations.

Comment: 13) Line 163: ‘well’ to be replaced by ‘good’

Response: Thanks very much for the reviewer’s kind remind. The word of ‘well’ in line 163 has been changed as ‘good’ in this revised manuscript.

Comment: 14) Regarding cytotoxic values, authors could mention the work by Safi et al in Nanotechnology 2010, 21, 145103 that shows that PAA-coated magnetic NPs with highly negative zeta potential are not internalized by biological cells (because of their negative membrane surface charge). Regarding PAA-coated samples, authors ascribe their high cytotoxicity to their positively charge: Actually this is only valid for –NH2 group that have high pKa value (around 9), because other week polycations like chitosan chains whose pKa is much lower (around 5-6) are non-toxic to cells.

Response: Thanks for the reviewer’s kind suggestion. The work by Safi et al in Nanotechnology 2010, 21, 145103 that shows that PAA-coated magnetic NPs with highly negative zeta potential are not internalized by biological cells (because of their negative membrane surface charge) have been mentioned in this revised manuscript. For PAH-coated samples, we agree with the reviewer that their high cytotoxicity should be attributed to the abundant amino groups that have high pKa value (around 9), rather than the positively charge of the PAH ligand. These have been corrected in this revised manuscript. The details are following.

For Fe3O4-PAA, the low cytotoxicity of our results is in good agreement with those reported in the literature, which showed that PAA-coated magnetic nanoparticles with highly negative zeta potential have good biocompatible and are not internalized by biological cells[54]. For Fe3O4-PAH, the large cytotoxicity may be attributed to the abundant amino groups that have high pKa value (around 9) of the PAH ligand on the surface of the Fe3O4 nanoparticles [48].

Comment: 15) Line 187 and Line 271: ‘Kg” to be replaced by ‘kg’

Response: Thanks for the reviewer’s kind remind. The words of ‘Kg’ in Line 187 and Line 271 have been replaced by ‘kg’ in this revised manuscript.

Comment: 16) Line 243: ‘Hydration’ to be replaced by ‘hydrodynamic’

Response: Thanks for the reviewer’s kind remind. The word of ‘Hydration’ in line 243 has been replaced by ‘hydrodynamic’ in this revised manuscript.

Comment: 17) Line 247: ‘nitration’ to be replaced by ‘dissolution in concentrated nitric acid’ (specify concentration)

Response: Thanks for the reviewer’s kind remind. The word of ‘Hydration’ in Line 247 has been replaced by ‘dissolution in concentrated nitric acid (15 mol/L) ’

Reviewer 2 Report

Authors report the synthesis and in vivo use of new T1-based nanoparticles. While results are potentially relevant there are many aspects that should be clarified: 

Line 41: “because the small Fe3O4 nanoparticles tend to form aggregation in vivo” I find this sentence too simple. Even if there are a few examples with such problems most of the reported examples show the contrary, see for example:

 (1)            Zhou, Z.; et al. Engineered Iron-Oxide-Based Nanoparticles as Enhanced T 1 Contrast Agents for Efficient Tumor Imaging. ACS Nano 2013, 7, 3287–3296.

(2)            Fernández-Barahona, I.et al. Cu-Doped Extremely Small Iron Oxide Nanoparticles with Large Longitudinal Relaxivity: One-Pot Synthesis and in Vivo Targeted Molecular Imaging. ACS Omega 2019, 4, 2719–2727.

(3)            Pellico, J.; et al One-Step Fast Synthesis of Nanoparticles for MRI: Coating Chemistry as the Key Variable Determining Positive or Negative Contrast. Langmuir 2017, 33, 10239–10247.

These references should be cited so the reader gets a clearer idea about the use of extremely small iron oxide nanoparticles as T1 agents.

Line 50: Authors should comment on the use of microwave chemistry for the synthesis of T1 nanoparticles.

Line 156: “The low r2/r1 ratio indicated that they are adequate as T1-weighted contrast agents.”  This sentence is a bit optimistic, in my opinion only Fe3O4-PAA can be considered a T1 contrast agent, the other two show too large r2 values to be useful in vivo for positive contrast, as in fact can be seen afterwards in the MRI experiments. This is particularly clear for Fe3O4-PVA, a r2/r1 ratio of 7.26 can’t ever be considered as T1 contrast agents. Author should clarify this since they just classify all of them as equal when they are not.

It is also noteworthy that these r1 values are obtained at 0.5 T, since it is well-known that these values will rapidly drop as the field increase they should measure the r1 at relevant fields like 3 T and/or 7 T or, at least 1.5 T.

Line 183: Authors injected 7.8 mg/Kg of the contrast agents, for a normal 30 g mouse, that is 0.234 mg that is about 2 times larger amount than previous published results. Authors should comment on this. With such large r1 values how is that they need to use that vast amount of iron?

Figure 5: If the most important aspect of these nanoparticles is their positive contrast why authors show image with false colours?, this is a legit way of showing images but strange when the most relevant result is the brightening of typical dark signals. They should show also the normal grey scale images so we can appreciate the brightening due to the accumulation of the particles.

Line 191: “…while the increased T1 signals of tumor was because the arrive but not seriously aggregation…” But, what is the reason, according to the authors, for this lack of aggregation in the tumor but not in the liver? If the particles accumulate due to the EPR effect why there is no aggregation??

As I mentioned before, the use of 0.5 T MRI machine is probably not the best idea since results are hardly applicable to normal human (3 T) or small animal (7 T) fields. If possible, authors should show r1 values at least at 1.5 T and images in relevant scanners. I know this is not always possible but results at 0.5 T lose a lot of relevance compared to other fields.

Author Response

Response to Reviewers

Reviewer 2:

Authors report the synthesis and in vivo use of new T1-based nanoparticles. While results are potentially relevant there are many aspects that should be clarified: 

Response: We really appreciate the reviewer for positive comments and constructive suggestions, which are very helpful for improving the quality of our manuscript.

Comment: 1) Line 41: “because the small Fe3O4 nanoparticles tend to form aggregation in vivo” I find this sentence too simple. Even if there are a few examples with such problems most of the reported examples show the contrary, see for example:

(1) Zhou, Z.; et al. Engineered Iron-Oxide-Based Nanoparticles as Enhanced T 1 Contrast Agents for Efficient Tumor Imaging. ACS Nano 2013, 7, 3287–3296.

(2) Fernández-Barahona, I.et al. Cu-Doped Extremely Small Iron Oxide Nanoparticles with Large Longitudinal Relaxivity: One-Pot Synthesis and in Vivo Targeted Molecular Imaging. ACS Omega 2019, 4, 2719–2727.

(3) Pellico, J.; et al One-Step Fast Synthesis of Nanoparticles for MRI: Coating Chemistry as the Key Variable Determining Positive or Negative Contrast. Langmuir 2017, 33, 10239–10247.

Response: Thanks for the reviewer’s kind remind. We agree with the reviewer that the sentence of “because the small Fe3O4 nanoparticles tend to form aggregation in vivo” is too simple. The dispersiveness of small Fe3O4 nanoparticles is generally associated with the ligands on surface of the nanoparticles. Generally, with appropriate ligands decorated on the surface, the Fe3O4 nanoparticles are able to exhibit low aggregation in vivo, and excellent T1-weight contrast. Therefore, great efforts have been devoted to explored surface ligands to improve the T1-performance of Fe3O4 nanoparticles. In this revised manuscript, the sentence of “Nevertheless, because the small Fe3O4 nanoparticles tend to form aggregation in vivo and subsequently display T2 rather than T1 contrast enhancement [14,15], the utilization of Fe3O4 nanoparticles as T1-weighted contrast agents still remains great challenge” has been replaced by “Nevertheless, small Fe3O4 nanoparticles that without appropriate ligands decorated on the surface tend to form aggregation and subsequently display T2 rather than T1 contrast enhancement [19, 20].”.

Nevertheless, small Fe3O4 nanoparticles that without appropriate ligands decorated on the surface tend to form aggregation and subsequently display T2 rather than T1 contrast enhancement [19, 20].

Comment: 2) These references should be cited so the reader gets a clearer idea about the use of extremely small iron oxide nanoparticles as T1 agents.

Response: Thanks for the reviewer’s suggestion. These three references have been cited in this revised manuscript as Ref. 4-6.

4.  Zhou, Z.; Wang, L.; Chi, X.; Bao, J.; Yang, L.; Zhao, W.; Chen, Z.; Wang, X.; Chen, X.; Gao, J. Engineered iron-oxide-based nanoparticles as enhanced T1 contrast agents for efficient tumor imaging. ACS Nano 2013, 7, 3287-3296.

5.  Fernández-Barahona, I.; Gutiérrez, L.; Veintemillas-Verdaguer, S.; Pellico, J.; Morales, M. d. P.; Catala, M.; del Pozo, M. A.; Ruiz-Cabello, J.; Herranz, F. Cu-doped extremely small iron oxide nanoparticles with large longitudinal relaxivity: one-pot synthesis and in vivo targeted molecular imaging. ACS Omega 2019, 4, 2719-2727.

6.  Pellico, J.; Ruiz-Cabello, J.; Fernández-Barahona, I.; Gutiérrez, L.; Lechuga-Vieco, A. V.; Enríquez, J. A.; Morales, M. P.; Herranz, F. One-step fast synthesis of nanoparticles for MRI: coating chemistry as the key variable determining positive or negative contrast. Langmuir 2017, 33, 10239-10247.

Comment: 3) Line 50: Authors should comment on the use of microwave chemistry for the synthesis of T1 nanoparticles.

Response: Thanks for the reviewer’s suggestion. The use of microwave chemistry for the synthesis of T1 nanoparticles has been commented in this revised manuscript. The details are following.

Besides, like the approach of microwave-assisted synthesis[6, 38], coprecipitation method is a procedure that can be easily scaled up [39].

Comment: 4) Line 156: “The low r2/r1 ratio indicated that they are adequate as T1-weighted contrast agents.”  This sentence is a bit optimistic, in my opinion only Fe3O4-PAA can be considered a T1 contrast agent, the other two show too large r2 values to be useful in vivo for positive contrast, as in fact can be seen afterwards in the MRI experiments. This is particularly clear for Fe3O4-PVA, a r2/r1 ratio of 7.26 can’t ever be considered as T1 contrast agents. Author should clarify this since they just classify all of them as equal when they are not.

Response: Thanks very much for the reviewer. We are sorry for our mistake. We agree with the reviewer that the Fe3O4-PAH and Fe3O4-PVA have too large r2 values, and are not suitable to be used in vivo for positive contrast. The sentence of “The low r2/r1 ratio indicated that they are adequate as T1-weighted contrast agents.” has been corrected as “For Fe3O4-PAA, the low r2 and r2/r1 ratio indicated that it is adequate as T1-weighted contrast agents.” in this revised manuscript.

The r2/r1 ratio for Fe3O4-PAA, Fe3O4-PAH, and Fe3O4-PVA were calculated to be 2.64, 3.82 and 7.27, respectively. For Fe3O4-PAA, the low r2 and r2/r1 ratio indicated that it is adequate as T1-weighted contrast agents.

Comment: 5) It is also noteworthy that these r1 values are obtained at 0.5 T, since it is well-known that these values will rapidly drop as the field increase they should measure the r1 at relevant fields like 3 T and/or 7 T or, at least 1.5 T.

Response: Thanks for the reviewer. We agree with the review that the r1 values will rapidly drop as the field increase (Chem. Mater. 2018, 30, 7950−7961). Due to the limited available devices and resources of our laboratory, it is challenge for us to measure the relaxivities at a higher field of 1.5 T or 3T. According to the reviewer’s suggestion, we re-measured the in vivo MR imaging on a 1T MRI scanner with a smaller intravenous injection dose of 1.3 mg Fe/kg body weight. The results revealed that both Fe3O4-PAA and Fe3O4-PVA were able to display remarkable brighten contrast enhancement for the mice tumor and kidney sites. The details are following.                                      

Figure 5. (a) T1-weighted MR images of mice collected before (control group) and after intravenous injection of Fe3O4-PAA at time points of 30, 60, 90, 120, 160 and 180 min. The corresponding relative T1-weighted signals extracted from (b) tumor (orange circle) and (c) kidney (dark yellow circle) sites.

Figure 6. (a) T1-weighted MR images of mice collected before (control group) and after intravenous injection of Fe3O4-PVA at time points of 40, 90, 120 and 180 min. The corresponding relative T1-weighted signals extracted from (b) tumor (orange circle) and (c) kidney (dark yellow circle) sites.

Comment: 6) Line 183: Authors injected 7.8 mg/Kg of the contrast agents, for a normal 30 g mouse, that is 0.234 mg that is about 2 times larger amount than previous published results. Authors should comment on this. With such large r1 values how is that they need to use that vast amount of iron?

Response: Thanks for the reviewer. We used the intravenous injection dose according to the reference (Nano Lett. 2016, 16, 2686−2691). We are sorry for our insufficiently consideration that the high dose of Fe3O4 nanoparticles may cause aggregation of the nanoparticle in vivo, especially in the liver and kidney sites. We re-measured the in vivo MR imaging on a 1T MRI scanner with a smaller intravenous injection dose of 1.3 mg Fe/kg body weight. The resulting MRI imaging showed higher imaging quality and spatial resolution, which revealed that both Fe3O4-PAA and Fe3O4-PVA were able to display remarkable brighten contrast enhancement for the mice tumor and kidney sites (The details are provided in the above Figure 5 and 6).

Comment: 7) Figure 5: If the most important aspect of these nanoparticles is their positive contrast why authors show image with false colours?, this is a legit way of showing images but strange when the most relevant result is the brightening of typical dark signals. They should show also the normal grey scale images so we can appreciate the brightening due to the accumulation of the particles.

Response: Thanks for the reviewer’s constructive suggestion. We agree with the reviewer that the grey scale images are the most effective way to show the images for appreciating the brightening that causing by the accumulation of the particles. We are very sorry for our insufficiently consideration to put the grey scale images in the supporting information. In this revised manuscript, the normal grey scale images have been provided in the main manuscript. The details are in the above Figure 5 and Figure 6.

Comment: 8) Line 191: “…while the increased T1 signals of tumor was because the arrive but not seriously aggregation…” But, what is the reason, according to the authors, for this lack of aggregation in the tumor but not in the liver? If the particles accumulate due to the EPR effect why there is no aggregation??

Response: Thanks very much for the reviewer. We feel maybe it is largely due to our unclear expression which causes confusion. Generally, for Fe3O4 nanoparticles, the aggregation phenomenon is more easily to occur with the increasing concentration. Liver is an organ with rich of reticuloendothelial-system, which can effectively sequester the administered materials, leading to relatively high concentration of Fe3O4 nanoparticles and negative contrast enhancement for this organ. This is a common phenomenon observed in references (ACS Appl. Nano Mater. 2018, 1, 894907). For the tumor site, the particles accumulated in this site due to the EPR effect. Generally, according to the reference (Nat Rev Mater. 2016; 1:16014), the amount of accumulated nanoparticles in tumor site is very limited, meaning that the concentration of Fe3O4 nanoparticles in tumor site is relatively low, and may have not reached the concentration for aggregation. In fact, in some case, Fe3O4 nanoparticles also exhibited aggregation in tumor site, leading to the T1-T2 switching contrast enhancement (Adv. Funct. Mater. 2018, 28, 1870221).

According to the reviewer’s suggestion, in this revised manuscript, the sentence of “…while the increased T1 signals of tumor was because the arrive but not seriously aggregation of Fe3O4-PAA nanoparticles, which instinctively featured T1 contrast enhancement in this site.” has been corrected as “The increased T1 signals of tumor and kidney can be attributed to the accumulation of Fe3O4-PAA nanoparticles that instinctively featured T1 contrast enhancement in these sites.”.

The increased T1 signals of tumor and kidney sites can be attributed to the accumulation of Fe3O4-PAA nanoparticles that instinctively featured T1 contrast enhancement in these sites.

Comment: 9) As I mentioned before, the use of 0.5 T MRI machine is probably not the best idea since results are hardly applicable to normal human (3 T) or small animal (7 T) fields. If possible, authors should show r1 values at least at 1.5 T and images in relevant scanners. I know this is not always possible but results at 0.5 T lose a lot of relevance compared to other fields.

Response: Thanks for the reviewer’s kind suggestion. We agree with the reviewer that the use of 0.5 T MRI machine is not the best idea since results and are hardly applicable to normal human (3 T) or small animal (7 T) fields. Due to the limited available devices and resources of our laboratory, it is difficult for us to obtain the r1 values at 1.5 T or 3T images in relevant scanners. According to the reviewer’s suggestion, we re-measured the in vivo MR imaging on a 1T MRI scanner with a smaller intravenous injection dose of 1.3 mg Fe/kg body weight. The results are showed in the above Figure 5 and Figure 6, which indeed have higher imaging quality and spatial resolution, as compared with that obtained on a 0.5 T MRI scanner. 

Round  2

Reviewer 1 Report

The authors have considered most of the Reviewer's remarks and corrected their manuscript accordingly. However there are still some minor defects that need to be corrected before publication:

1) Figure S1 has still too many digits in the provided TEM diameters. The displayed values should rather be 4.5+/-0.8 nm, 7.4+/-1.3 nm, and 2.8+/-0.4 nm.

2) Figure S2 legend: replace 'solution' by 'suspension'

3) The answer provided by authors about molar mass or molecular weight (usual abbreviation Mw instead of M.w.) show that the 3 polymers also differ by their lenghts. Therefore the Reviewer suggest slightly modifying the title of the paper into 'Macromolecules with Different Charges, Lengths, and Coordination Groups for Coprecipitation Synthesis of Magnetic Iron Oxide Nanoparticles as T1 MRI Contrast Agents'

4) Apparently the authors made novel in vivo MRI experiments at a higher magnetic field (1T instead of 0.5 T in the previous version of the manuscript), according to the the two Reviewers' recommendation (to be closer to a human clinical MRI). However Materials & Methods part 3.3 still contains description of a 0.5 T (NMI20) MRI scanner: Please correct this section accordingly (and precise the value of B0 used in the legends of Figures 5 and 6).
5) Regarding relaxometric measurements (values still done at 0.5 T), the Reviewer had asked to specify the R1° and R2° values of pure water (or buffer solution). Unlike what authors claim to have corrected, the two lines are still superimposed at [Fe]=0 mM, which is not possible because T1° and T2° of water are not equal (for example T1°=4 s and T2°=0.85 s are typical values measured at 1.4T and 37°C).  A plausible explanation is that the plots of Fig3 represent R1-R1° and R2-R2° versus concentration rather than R1 and R2 as stated in the legend (in that case it has to be clearly written, and the R1° and R2° values need to be displayed in the legend).

It will be possible to publish the article in Nanomaterials when the authors make these final corrections to their manuscript.

Author Response

Response to Reviewers

Reviewer 1:

Comment: The authors have considered most of the Reviewer's remarks and corrected their manuscript accordingly. However there are still some minor defects that need to be corrected before publication:

Response: We really appreciate the reviewer for positive comments and constructive suggestions, which are very helpful for improving the quality of our manuscript.

Comment: 1) Figure S1 has still too many digits in the provided TEM diameters. The displayed values should rather be 4.5+/-0.8 nm, 7.4+/-1.3 nm, and 2.8+/-0.4 nm.

Response: Thanks very much for the reviewer. We are sorry for our mistake. The displayed values in Figure S1 have been modified as “4.5+/-0.8 nm, 7.4+/-1.3 nm, and 2.8+/-0.4 nm” in this revised manuscript.                                      

Figure S1. Particle size distribution of (a) Fe3O4-PAA, (b) Fe3O4-PAH and (c) Fe3O4-PVA acquired from the TEM images.

Comment: 2) Figure S2 legend: replace 'solution' by 'suspension'

Response: Thanks for the reviewer’s kind remind. The word of ‘solution’ in the legend of Figure S2 has been changed by ‘suspension’ in this revised manuscript.

Comment: 3) The answer provided by authors about molar mass or molecular weight (usual abbreviation Mw instead of M.w.) show that the 3 polymers also differ by their lenghts. Therefore the Reviewer suggest slightly modifying the title of the paper into 'Macromolecules with Different Charges, Lengths, and Coordination Groups for Coprecipitation Synthesis of Magnetic Iron Oxide Nanoparticles as T1 MRI Contrast Agents'

Response: Thanks very much for the reviewer’s kind suggestion. In this revised manuscript, the abbreviation has been changed as “Mw”, and the paper title has been modified into 'Macromolecules with Different Charges, Lengths, and Coordination Groups for Coprecipitation Synthesis of Magnetic Iron Oxide Nanoparticles as T1 MRI Contrast Agents'.

Comment: 4) Apparently the authors made novel in vivo MRI experiments at a higher magnetic field (1T instead of 0.5 T in the previous version of the manuscript), according to the the two Reviewers' recommendation (to be closer to a human clinical MRI). However Materials & Methods part 3.3 still contains description of a 0.5 T (NMI20) MRI scanner: Please correct this section accordingly (and precise the value of B0 used in the legends of Figures 5 and 6).

Response: Thanks for the reviewer’s kind remind. We feel maybe it is largely due to our unclear expression which causes confusion. The in vitro and in vivo MRI experiments were carried out on 0.5 T and 1T MRI scanners, respectively. In Materials & Methods part 3.3, the “0.5 T (NMI20) MRI scanner” was the scanner for the in vitro experiments. The 1T MRI scanner for in vivo experiment has been descripted in Materials & Methods part 4.5.

The value of B0 = 1 T has been provided in the legends of Figure 5 and 6 in this revised manuscript.

Comment: 5) Regarding relaxometric measurements (values still done at 0.5 T), the Reviewer had asked to specify the R1° and R2° values of pure water (or buffer solution). Unlike what authors claim to have corrected, the two lines are still superimposed at [Fe] = 0 mM, which is not possible because T1° and T2° of water are not equal (for example T1°=4 s and T2°=0.85 s are typical values measured at 1.4T and 37 °C).  A plausible explanation is that the plots of Fig3 represent R1-R1° and R2-R2° versus concentration rather than R1 and R2 as stated in the legend (in that case it has to be clearly written, and the R1° and R2° values need to be displayed in the legend).

Response: Thanks very much for the reviewer. We agree with the reviewer that the two lines of 1/T1 and 1/T2 at [Fe] = 0 mM should not be superimposed. We feel that maybe it is due to the two values of 1/T1 and 1/T2 for the pure water are too small, which making the two lines of 1/T1 and 1/T2 seem likely superimposed at [Fe] = 0 mM. According to the reviewer’s suggestion, the R1° and R2° values of pure water have been provided in the legend of Figure 3 in this revised manuscript.

Figure 3. Plot of 1/T against Fe concentration for (a) Fe3O4-PAA, (b) Fe3O4-PAH, and (c) Fe3O4-PVA nanoparticles. The values of r1and r2 were calculated based on the slopes of the corresponding fitting lines (B0 = 1 T, 25 oC). The T1° and T2° values of pure water were 2.79 and 3.01 s, respectively. (d) T1-weight MR images of the aqueous dispersion of Fe3O4-PAA, Fe3O4-PAH, and Fe3O4-PVA nanoparticles with different Fe concentrations.

Reviewer 2 Report

All my concerns have been addressed. 

thanks

Author Response

Thanks very much for the reviewer.